# miR551b Regulates Colorectal Cancer Progression by Targeting the ZEB1 Signaling Axis

**DOI:** 10.3390/cancers11050735

**Published:** 2019-05-27

**Authors:** Kwang Seock Kim, Dongjun Jeong, Ita Novita Sari, Yoseph Toni Wijaya, Nayoung Jun, Sanghyun Lee, Ying-Gui Yang, Sae Hwan Lee, Hyog Young Kwon

**Affiliations:** 1Soonchunhyang Institute of Medi-bio Science (SIMS), Soonchunhyang University, Cheonan-si 31151, Korea; kimks5005gt@gmail.com (K.S.K.); itanov07@gmail.com (I.N.S.); yoseph.toni@gmail.com (Y.T.W.); jny0407@naver.com (N.J.); october-92@daum.net (S.L.); syyangyinggui@163.com (Y.-G.Y.); 2Department of Pathology, College of Medicine, Soonchunhyang University, Cheonan-si 31151, Korea; juny1024@sch.ac.kr; 3Soonchunhyang Medical Science Research Institute, College of Medicine, Soonchunhyang University, Cheonan-si 31151, Korea; 4Liver Clinic, Soonchunhyang University Cheonan Hospital, Cheonan-si 31151, Korea

**Keywords:** miR551b, colorectal cancer, ZEB1, EMT

## Abstract

Our current understanding of the role of microRNA 551b (miR551b) in the progression of colorectal cancer (CRC) remains limited. Here, studies using both ectopic expression of miR551b and miR551b mimics revealed that miR551b exerts a tumor suppressive effect in CRC cells. Specifically, miR551b was significantly downregulated in both patient-derived CRC tissues and CRC cell lines compared to normal tissues and non-cancer cell lines. Also, miR551b significantly inhibited the motility of CRC cells in vitro, including migration, invasion, and wound healing rates, but did not affect cell proliferation. Mechanistically, miR551b targets and inhibits the expression of ZEB1 (Zinc finger E-box-binding homeobox 1), resulting in the dysregulation of EMT (epithelial-mesenchymal transition) signatures. More importantly, miR551b overexpression was found to reduce the tumor size in a xenograft model of CRC cells in vivo. Furthermore, bioinformatic analyses showed that miR551b expression levels were markedly downregulated in the advanced-stage CRC tissues compared to normal tissues, and ZEB1 was associated with the disease progression in CRC patients. Our findings indicated that miR551b could serve as a potential diagnostic biomarker and could be utilized to improve the therapeutic outcomes of CRC patients.

## 1. Introduction

Colorectal cancer (CRC) is the third most commonly diagnosed cancer and second leading cause of cancer-related deaths worldwide, accounting for approximately 1.8 million new cases and about 881,000 deaths for both sexes in 2018 [1]. Despite the technological advancements in screening strategies and therapeutic regimens, CRC has been associated with poor prognosis, especially when metastasis to the lymph nodes or distant organs has occurred, and almost one-third of CRC patients will ultimately die as a result of metastasis [2,3]. Therefore, there is an urgent need to elucidate the mechanisms underlying metastasis in CRC, which could lead to the development of novel therapeutics and improve the management of advanced CRC cases [4]. One of the key molecular steps in the process of distant metastasis is the epithelial-to-mesenchymal transition (EMT) [5,6]. The EMT is a complex process that involves the dissolution of cell–cell junctions and loss of apico-basolateral polarity, eventually leading to the formation of migratory mesenchymal cells with invasive properties [7]. The EMT permits invasion and emigration of various cancers and is closely associated with a poor prognosis in CRC [8,9,10].

MicroRNAs (miRNAs) are small non-coding RNAs that post-transcriptionally regulate gene expression and play important roles in various physiological processes [11,12]. miRNAs normally contain 19–25 nucleotides and are complementary to the non-coding regions of mRNA 3′ ends of their target mRNAs, thereby inhibiting the translation [13,14]. Previous studies indicated that miRNAs can influence the expression of up to 30% of all genes and thus potentially regulate thousands of genes [7,15]. Most importantly, miRNAs regulate various stages of tumor development and progression, including proliferation, migration, and multi-drug resistance [16,17,18]. Previous studies indicated that miR551b can act as either an oncogenic or a tumor suppressive factor. As an oncogenic factor, miR551b-3p was shown to upregulate STAT3 (Signal transducer and activator of transcription 3) protein levels, contributing to resistance to apoptosis, higher rates of survival, and increased cell proliferation in ovarian cancer [19]. In addition, miR551b-3p targets GLIPR2 (GLI Pathogenesis Related 2) to promote tumor growth in high-risk head and neck cancer (HRHNC) by modulating autophagy [20,21]. By contrast, miR551b-3p expression levels were found to be inversely correlated with the prognosis of gastric cancer (GC) as a result of inhibiting ERBB4 expression [22,23]. However, the role of miR551b in the pathogenesis of CRC remains unknown. Here, we identified the tumor suppressive effects of miR551b in CRC and the molecular mechanisms underlying these effects.

## 2. Materials and Methods

### 2.1. Cell Lines

Human colorectal cancer cell lines SW620 and HCT116 were purchased from the Korean Cell Line Bank (KCLB, Seoul, Korea). HCT116 cells were grown in DMEM medium (Corning, St. Louis, MO, USA), and SW620 cells were grown in RPMI-1640 medium (Corning) supplemented with 10% FBS (Corning), 1% MEM essential amino acids (Corning), and 1% penicillin/streptomycin (Gibco, Grand Island, NY, USA) at 37 °C in a humidified atmosphere with 5% CO_2_.

### 2.2. miR551b Construct and Infection

The pri-miR551b and the 490-bp flanking region was PCR amplified from human genomic DNA using specific primers (forward, 5′-GGCGAACTCGAGTCTGCCAGATGTGCTCTCCT-3′ and reverse, 5′-GGCGCCGAATTCTTTTTCTCTGGAAGTCCTGCAT-3′) and subsequently cloned into lentiviral pZeo vector as previously described [24]. 293T cells were transfected with the viral construct along with pMD2 and psPAX constructs to generate viruses using the iN-fectTM in vitro transfection reagents (iNtRON, Seongnam-si, Korea) and following the manufacturer’s protocol. Viral supernatants were collected on days 2 and 3 after transfection and were used to infect the target cells. After lentiviral infection, GFP positive cells were sorted by FACS Aria III (BD Bioscience, Sandy, UT, USA) and used for experiments.

### 2.3. Transfection with miR-551b Mimics

SW620 and HCT116 cells (6 × 10^6^ cells per well) were grown overnight in a 10-cm plate, followed by transfection with either negative control, miR551b-3p mimic, or miR551b-5p mimic (10 μM) according to the manufacturer’s instructions. The miR551b-3p mimic, miR551b-5p mimic, and negative control were purchased from Bioneer Co. Ltd (Dajun, Korea) for all experiments.

### 2.4. Cell Proliferation Assay (MTT Assay)

MTT assay was performed to evaluate cell proliferation by using Cell Proliferation Kit I according to the manufacturer’s instructions (Roche, Basel, Switzerland). Briefly, cells (5 × 10^3^ cells) were seeded into a 96-well plate and incubated for an additional 72 h. Cells were incubated with 5 mg/mL MTT solution for 4 h and then solubilized with 100 µL of solubilization solution (10% SDS in 0.01 M HCl) overnight. Absorbance was read at 575 nm and 650 nm using a plate reader.

### 2.5. Migration and Invasion Assay

The transwell insert (8 μm pore size) system (Corning, USA) coated with or without 20 μL of Matrigel (BD Bioscience) were respectively used to examine the cell invasion and migration in vitro as shown before [25]. The culture insert was attached on bottom of a 24-well plate, and 100 μL of serum-free media containing 1.5 × 10^5^ cells were seeded onto each well of the insert. Media containing 10% FBS (500 μL) was added outside the transwell culture insert. Cells were incubated at 37 °C for 24 h in a humidified atmosphere with 5% CO_2_ for evaluating cell migration and invasion. Transwells were washed twice with PBS and cleaned using cotton swaps. The cells were fixed with 1% formaldehyde for 15 min, washed twice with PBS, stained with 0.1% of crystal violet for 15 min, and then observed using a microscope (Leica, Wetzlar, Germany).

### 2.6. Wound Healing Assay

The cells (1 × 10^5^ cells) were seeded onto 24-well plates at 24 h before treatment. Next, the plates were scratched using the end of a 200-µL pipette tip (0 h time point), and cells were washed twice with PBS to remove the loose cells. Later, the cells were cultured for 2 days, and images were captured every 24 h to assess the migration rates in each group of transfected cells.

### 2.7. RNA Extraction and Real Time qPCR (RT-qPCR)

RNA was isolated using Trizol (Invitrogen, Carlsbad, CA, USA), miR isolation kit (Qiagen, Hilden, Germany), or Hybrid R (Gene All, Seoul, Korea), and subsequently converted to cDNA using ReverTra Ace^®^ qPCR Kit (Toyobo, Osaka, Japan) or miRCURY LNATM Universal cDNA synthesis Kit (Qiagen) according to the manufacturer’s instructions. To determine the level of gene expression, RT-qPCR was performed using the qPCR Master Mix Kit (Toyobo) or miRCURY LNATM SYBR^®^ Green PCR Kit (Qiagen). Primer sequences used for RT-qPCR were shown in Appendix A. The primers of miR551b-3p, miR551b-5p and U6 (control) for all RT-qPCR experiments were purchased from Qiagen.

### 2.8. Western Blotting Analysis

Cell lysates were harvested using RIPA lysis buffer for 30 min on ice and centrifuged at 13,000 rpm for 10 min at 4 °C. The protein concentration of each supernatant was determined by Bio-Rad Protein Assay (Bio-Rad Laboratories, Inc., Hercules, CA, USA). An equal amount of each protein extract (30 μg) was resolved using 10% polyacrylamide gel and electro-transferred onto a 0.45-μm hybridization nitrocellulose filter (HATF) membrane (MilliporeBurlington, MA, USA) using Trans-blot Turbo (Bio-Rad Laboratories, Inc.). Membranes were immunoblotted with either goat polyclonal anti-actin antibody (1:3000 dilution, Abcam, Cambridge, UK), and rabbit monoclonal anti-ZEB1 antibody (1:2000 dilution, Cell Signaling, Danvers, MA, USA) overnight at 4 °C. Next, membranes were incubated with either HRP-conjugated anti-rabbit immunoglobulin (Cell Signaling) or HRP-linked anti-goat immunoglobulin (1:3000 dilution, Santa Cruz Biotechnology, Santa Cruz, CA, USA) for 1 h at room temperature. Protein signals were detected by enhanced chemiluminescence (Thermo, Waltham, MA, USA) using the Amersham Imager 600 (GE Healthcare Life Sciences, Chicago, IL, USA). The images of whole membranes were shown in Appendix A.

### 2.9. Dual Luciferase Reporter Assay

miRNA binding sites located in the 3′-UTR of ZEB1 were predicted using the miRanda tool available at http://www.microrna.org. The putative binding site by miR551b was 5′-CUAGCAUUUGUUGAUUU-3′. The ZEB1 luciferase reporter construct was generated by amplifying the human ZEB1 mRNA 3′-UTR sequence by PCR (huZEB1 forward, 5′- GGCGCAGAATTCATGAAAGTTACAAATTATAATACTGTG-3′ and reverse, 5′-GGCGCAGGATC CCGGGCTTCATTTG TCTTTTCTTCAGAC-3′) and subsequently cloned into the psiCHECK-2 vector. The plasmid was then co-transfected with either control vector or miR551b overexpression into 293T cells using Lipofectamine (iNtRON Biotechnology, Seongnam-si, Korea) according to the manufacturer’s instructions. Luciferase activity was measured by Dual-Luciferase Assay Kit (Promega, Madison, WI, USA) at 24 h after transfection.

### 2.10. Xenograft Mouse Model

To examine the effect of miR551b on the growth of tumor cells in vivo, SW620 cells were virally transduced with either pZeo vector or miR551b overexpression construct. The transduced cells (3 × 10^6^ cells/mouse) were then subcutaneously injected into NSG mice (NOD.Cg-*Prkdc^scid^ Il2rg^tm1Wjl^*/SzJ, The Jackson Laboratory, Bar Harbor, ME, USA), and the tumor growth/ volume was monitored every 3 days for 1 month. Tumors were measured with an electronic caliper (Fowler Sylvac UltraCal Mark III, Sylvac, Lausanne, Switzerland). The average tumor volume was calculated using the formula TV (mm3) = πW^2^ × L/6, where W is the smaller diameter and L is the larger diameter. The tumor tissues collected from sacrificed mice were processed for morphological analysis and molecular analysis. All animal experiments were performed according to protocols approved by Soonchunhyang University Institutional Animal Care and Use Committee (2019-0025).

### 2.11. Microarray Data Analysis

MicroRNA datasets GSE81582, GSE41655, GSE18392, GSE98406, and GSE30454 were downloaded from the Gene Expression Omnibus database (https://www.ncbi.nlm.nih.gov/geo/). A total of 64 normal control samples and 208 CRC samples were collected from these datasets and used for analysis with GEO2R (https://www.ncbi.nlm.nih.gov/geo/geo2r/) to obtain the differential expression of has-miR-551b between normal and CRC tissue samples. MicroRNA dataset GSE41655 was analyzed to determine the correlation between hsa-miR-551b expression and tumor progression.

### 2.12. Survival Analysis of ZEB1

The Kaplan-Meier plotter tool in R2: Kaplan Meier Scanner (https://r2.amc.nl) was utilized to obtain the correlation of ZEB1 expression with survival rate using Tumor Colon CIT (Combat)—Marisa dataset (GSE39582). Patients were categorized into 2 groups; the top 25% (high expression) and the rest of patients (low expression) depending on the expression of ZEB1 (high expression, *n* = 140; low expression, *n* = 417).

### 2.13. Information of Tissue Specimens

For miRNA qRT-PCR experiments, the 10 pair tissues were obtained from colorectal carcinoma patients who underwent surgery at Soonchunhyang University Cheonan Hospital in Korea. All clinicopathological data including age, gender, TNM stage, vascular invasion, Lymphatic invasion, and perineural invasion were noted in Appendix A. This study and clinicopathological data were approved by the Ethics Committee of Soonchunhyang University, Cheonan Hospital (2018-07-061-003).

### 2.14. Statistical Analysis

Results of RT-qPCR, western blotting, migration, invasion, wound healing, and hsa-miR-551b expression were analyzed with Student’s *t*-test. Data integration and statistical analysis of microRNA datasets were conducted with GraphPad Prism 6.0 (GraphPad Inc., San Diego, CA, USA). The association of miR551b between normal and CRC progression was compared using one-way ANOVA. A *p*-value of less than 0.05 was considered statistically significant in all assessments.

## 3. Results

### 3.1. miR551b-3p is Downregulated in CRC Cell Lines and Patient-Derived Samples

To understand whether miR551b exhibits an oncogenic or tumor suppressive effect, we first determined the expression levels of two miR551b variants, namely, miR551b-3p and miR551b-5p, in non-CRC cell lines (CCD84, and CCD18co), non-metastatic CRC cell lines (CoLo201, SW480), and metastatic CRC cell lines (HCT116, SW620, and HT29). Interestingly, miR551b-3p levels were considerably downregulated in CRC cell lines relative to those in non-CRC cell lines (Figure 1A). However, miR551b-5p expression levels were very low and were not differentially expressed between non-CRC cell lines and CRC cell lines (Figure 1A). To determine the involvement of each variant in patient-derived samples, we analyzed miR551b-3p and miR551b-5p levels in both patient-derived CRC samples and the corresponding normal tissues. Consistent with the findings obtained from the cell lines, miR551b-3p levels were markedly lower in CRC samples compared to those in normal samples (Figure 1B). In addition, CRC samples and normal samples showed no significant differences in miR551b-5p levels (Figure 1B). The above findings indicated that miR551b-3p was downregulated in CRC cells compared to normal cells, suggesting that miR551b-3p is involved in the development and progression of CRC.

### 3.2. Ectopic Expression of miR551b Inhibits the Migration and Invasion of CRC Cell Lines In Vitro

Given the potential link between miR551b-3p and CRC progression, we determined the functional effects of miR551-3p on CRC cells. We overexpressed miR551b virally in two CRC cell lines, namely, SW620 and HCT116, and determined the corresponding functional consequences, including proliferation and motility of CRC cells. RT-qPCR analysis was performed to confirm the overexpression of miR551b. As shown, miR551b-3p levels were markedly higher in miR551b-transduced cells compared to those in pZeo vector-transduced cells, but miR551b-5p was almost not detected (Figure 2A,B). To evaluate whether miR551b plays a role in the growth of CRC cell lines, we analyzed the effect of miR551b on cell proliferation by MTT assay. The results revealed no significant differences in cell proliferation between miR551b-transduced cells and control cells (Figure 2C,D). Previous studies showed that miR551b mimic inhibited the migration and invasion of gastric cancer cell lines [23]. Thus, we investigated whether miR551b overexpression exerted similar inhibitory effects on the motility of CRC cells. To investigate this, we seeded control vector or miR551b-transduced cells on a transwell system and assayed the migration of CRC cells. miR551b-transduced SW620 and HCT116 cells showed 47% and 53% lower migration rates relative to the control cells, respectively (Figure 2E–G). Thus, we examined the invasion of CRC cells using a matrigel-coated insert of a transwell system. As shown in Figure 2H–J, miR551b-transduced SW620 and HCT116 cells exhibited 40% and 38% lower cell invasion rates relative to the control-transduced cells, respectively. To confirm the inhibitory effect of miR551b on the motility of CRC cells using an independent approach, scratch-induced wound healing assay was performed. Control vector- or miR551b-transduced SW620 and HCT116 cells were scratched and monitored for 2 days to evaluate the wound healing rates. miR551b-transduced cells exhibited a significantly slower rate of healing rates compared to those of control cells (Figure 2K–N). Collectively, these findings indicated that miR551b exerted an inhibitory effect on the motility of CRC cells.

### 3.3. miR551b-3p Mimic Inhibits Motility of CRC Cell Lines In Vitro

Ectopic expression of miR551b inhibited the motility of CRC cells, and these results were independently verified using miR551b mimics. Considering that there are two miR551b variants, namely, miR551b-5p and miR551b-3p, we determined which form is functionally important. First, SW620 cells were transiently transfected with either control, miR551b-5p mimic, or miR551b-3p mimic; the expression levels of miR551b variants were confirmed by RT-qPCR (Figure 3A,B). Next, functional assays, including proliferation and motility assays, were performed. Consistent with the results obtained by ectopic expression of miR551b, cells transduced with miR551b-5p mimic or miR551b-3p mimic showed comparable proliferation rates to those of control cells (Figure 3C,D). Interestingly, miR551-3p mimic, but not miR551b-5p mimic, significantly decreased the cell migration, invasion, and wound healing rates of SW620 cells compared to those of control cells (Figure 3E–N).

Thus, the above results suggested that the inhibitory effect of miR551b on the motility of CRC cells was associated with miR551b-3p, but not with miR551b-5p.

### 3.4. miR551b Targets ZEB1, Leading to the Dysregulation of the EMT Signaling Pathway

Given that both ectopic expression of miR551b and miR551b-3p mimic treatment exerted the inhibitory effects on the motility of CRC cells, we attempted to identify the targets of miR551b. To determine the putative targets of miR551b, several web-based tools, such as miRanda and Targetscan, were utilized, and the prediction identified the 3′-UTR of ZEB1 as a potential target of miR551b (Figure 4A). Dual-luciferase reporter assay was performed to evaluate whether miR551b directly binds to the 3′-UTR of ZEB1. Luciferase reporter assay was then performed to determine the binding of miR551b on the ZEB1 UTR. Cells co-transfected with miR551b overexpression and ZEB1 UTR showed a significantly lower (50%) luciferase signal compared to those transfected with control vector and ZEB1 UTR (Figure 4B), suggesting that miR551 directly binds to the ZEB1 UTR, leading to reduced luciferase activity. Thus, we determined whether ZEB1 levels were altered by miR551b overexpression in SW620 cells by RT-qPCR and immunoblotting. Consistent with the results of luciferase assay, both RNA and protein levels of ZEB1 were significantly downregulated by ectopic expression of miR551b relative to those in control cells (Figure 4C–E). These results suggested that miR551b directly targets the 3′UTR of ZEB1, resulting in the downregulation of ZEB1 levels. ZEB1 is well-known as a transcription factor that promotes tumor invasion and metastasis by inducing the EMT in carcinoma cells [26]. Therefore, we performed RT-qPCR to determine the effect of miR551b on the expression of the EMT genes and understand the signaling pathway downstream of ZEB1.

Epithelial gene *E-CAD* level was upregulated following miR551b overexpression, whereas the levels of mesenchymal genes, such as *N-CAD*, *SNAIL*, and *VIMENTIN*, were downregulated by miR551b overexpression (Figure 4F–I), suggesting that miR551b overexpression leads to the dysregulation of EMT signatures in CRC cells by targeting ZEB1. These data were consistent with the previous reports, showing that ZEB1 is implicated in EMT by repressing E-CAD [26,27,28,29].

### 3.5. miR551b Overexpression Suppressed Tumor Growth in a Mouse Xenograft Model of CRC Cells In Vivo

The above findings showed that miR551b overexpression suppressed the EMT characteristics of CRC cells in vitro. Therefore, we determined the in vivo relevance of miR551b using a mouse xenograft model. SW620 cells were transduced virally with control vector or miR551b and then subcutaneously injected into immune-compromised Nod Scid gamma (NSG) mice. The tumor sizes were measured every 3 days and analyzed at 1 month after transplantation. The tumors that developed from miR551b-transduced SW620 cells were considerably smaller than the control tumors (pZeo, 1307.9 ± 152.4 mm^3^ vs. miR551b, 754.3 ± 122.1 mm^3^; Figure 5A,B). Expectedly, miR551b-3p expression level was significantly higher in miR551b-transduced tumors compared to those in the control tumor, but miR551b-5p was not detected in both miR551b- and control-transduced tumors (Figure 5C). The weights of miR551b-transduced tumor were 1.7-fold lower compared to those of the control tumor (Figure 5D). The above findings suggested that ectopic expression of miR551b significantly suppressed tumor growth in a mouse xenograft model of CRC cells in vivo.

### 3.6. The Inverse Correlation Between miR551b and ZEB1 in the Prognosis of CRC 

Our results showed that miR551b exerted a tumor suppressive effect in CRC cells in vitro and in vivo. Therefore, we tested the relevance of miR551b in patient prognosis. Using publicly available datasets, we reanalyzed the expression of miR551b and its target, ZEB1, in patient-derived samples. We collected a total of 787 CRC, 42 dysplasia and 79 normal control samples from six datasets (GSE81582, GSE41655, GSE18392, GSE98406, GSE30454, and GSE39582). Interestingly, we found that miR551b expression levels were considerably downregulated in advanced-stage tumors compared to those in normal tissues, indicating that miR551b levels are inversely correlated with the prognosis of CRC (Figure 6A,B). In addition, CRC patients were categorized into two groups, depending on ZEB1 expression levels (high or low expression of ZEB1). In particular, higher ZEB1 levels were closely associated with a poor prognosis of CRC patients (Figure 6C), suggesting that the inverse relationship between miR551b and ZEB1 could influence disease progression of CRC patients.

## 4. Discussion

Nowadays, miRNAs are considered important factors affecting cancer progression and disease recurrence. miRNAs have been implicated in various cellular processes, including proliferation, translation, and the EMT. The functions of miR551b have been debated because it can act as an oncogenic factor in OC (Ovarian cancer) and HRHNC or as a tumor suppressor in GC (Gastric cancer) [19,21,22,23]. However, the significance of mir551b in CRC remains unknown. Herein, using both ectopic expression of miR551b and miR551b mimics, we showed that miR551b exerts a tumor suppressive effect in CRC cells. miR551b-3p was significantly downregulated in both tumor tissues and CRC cell lines. In addition, both miR551b overexpression and treatment with miR551b-3p mimic significantly reduced the migration, invasion, and wound healing rates of CRC cells. Notably, miR551b overexpression reduced the tumor sizes using xenografts of CRC cells in vivo. Furthermore, results from both RT-qPCR and bioinformatic analyses using publicly available datasets showed that miR551b is downregulated in tumor samples relative to normal tissues. Consistent with our findings, Song et al., recently demonstrated that miR551b was significantly downregulated in the GC tumor tissues compared to the normal stomach tissues and that GC prognosis was inversely correlated with miR551b expression [23]. However, miR551b appeared to have a different role in OC, in which miR551b expression was found to be positively correlated with progression of OC by promoting tumor growth [21]. Thus, it appears that the function of miR551b may be finely regulated depending on genetic context and tumor type. We observed differences in miR551b-mediated effects between OC and CRC. In OC, miR551b binds to the promoter of STAT3, leading to activation of the transcription in OC [21]. However, we showed that miR551b binds to the 3′UTR of ZEB1 in CRC, leading to the downregulation of ZEB1 mRNA and protein levels. In addition, we did not exclude the possibility that miR551b activates the promoters of some target genes. Thus, the differential regulation and functional differences associated with miR551b between CRC and OC should be addressed in detail in subsequent studies.

miR551b has two variants, namely, miR551b-3p and miR551b-5p. We analyzed the expression and functional difference between the two variants. Results revealed that miR551b-3p was more strongly expressed in both normal tissues and non-cancer cell lines, but showed weaker expression in patient-derived CRC samples and CRC cell lines. Contrastingly, normal and cancer tissues showed no significant differences in miR551b-5p expression. In addition, miR551-3p mimic, but not miR551-5p, exerted a tumor-suppressive effect. Thus, we expected that the tumor-suppressive functions of miR551b in CRC cells were mainly mediated by miR551b-3p. Several miRNAs, such as miR200 [30], miR1199-5p [31], and miR128-3p [32], were shown to target ZEB1, leading to the coordination of EMT. Interestingly, we also identified ZEB1 as a miR551b target. By recruiting co-suppressors or co-activators, ZEB1 was shown to either downregulate or upregulate the expression of target genes and promote tumor metastasis via EMT induction in cancer cells [26,28,33]. We showed that ZEB1 and EMT-related signatures, such as E-CAD, N-CAD, SNAIL, and VIMENTIN, were dysregulated by miR551b expression. Thus, the finding that ZEB1 is regulated by miR551b has a huge potential impact on research on the EMT and cancer metastasis and suggested that miR551b is a good therapeutic alternative for inhibiting the EMT and tumor progression in CRC.

## 5. Conclusions

Collectively, our findings provided important clues underlying the mechanisms by which miR551b mediates the EMT and tumor progression in CRC. Our results showed that miR551b regulates the EMT characteristics of CRC cells by targeting ZEB1 and highlighted the potential diagnostic and prognostic values of miR551b as a biomarker.

## Figures and Tables

**Figure 1 cancers-11-00735-f001:**
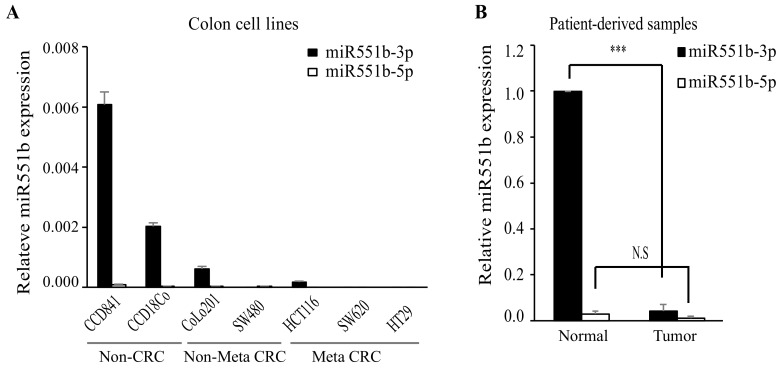
**miR551b-3p is downregulated in CRC patient-derived samples and CRC cell lines.** (**A**) and (**B**), The expression levels of miR551b-3p and miR551b-5p were determined by RT-qPCR in CRC cell lines (**A**) and patient-derived samples (*n* = 10; (**B**)). Data were shown as mean ± SEM of three independent experiments. *** *p* < 0.001. N.S means not significant.

**Figure 2 cancers-11-00735-f002:**
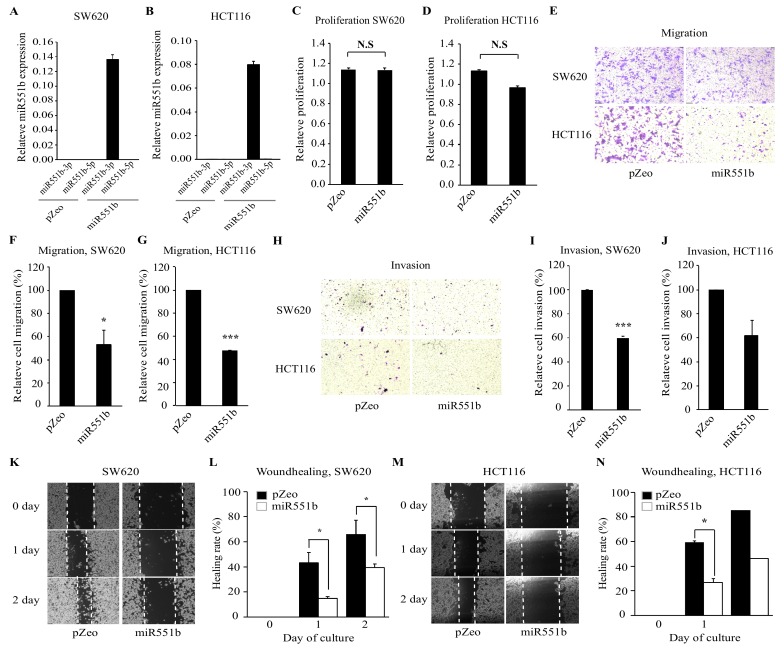
**miR551b****overexpression suppresses the motility of CRC cell lines.** (**A** and **B**), The expression levels of miR551b-3p and miR551b-5p were determined by RT-qPCR in SW620 (**A**) and HCT116 (**B**). (**C** and **D**), SW620 and HCT116 cells were transduced with either control vector (pZeo) or miR551b, and subsequently incubated for 72 h to determine cell proliferation rates by the MTT assay. N.S means not significant. (**E**–**J**), Control (pZeo) or miR551b-transduced CRC cell lines were seeded in a transwell non-coated or coated with Matrigel, followed by incubation for 24 h for evaluation of cell migration and invasion, respectively. Cells that had migrated to the lower surface of the transwell were stained and quantified. Images were taken using an inverted microscope and representative images were shown (**E**, migration; **H**, invasion). Data were presented as mean ± SEM of three independent experiments (**E**–**G**, migration; **H**–**J**, invasion). * *p* < 0.05, *** *p* < 0.001. (**K**–**N**), Control (pZeo) or miR551b-transduced SW620 (**K** and **L**) and HCT116 (**M** and **N**) cells were seeded and analyzed by in vitro scratch assays. Images were captured at 0, 1, and 2 days after incubation. The dotted lines defined the areas lacking cells. Representative images were shown (**K** and **M**). Data were shown as mean ± SEM of three independent experiments. * *p* < 0.05.

**Figure 3 cancers-11-00735-f003:**
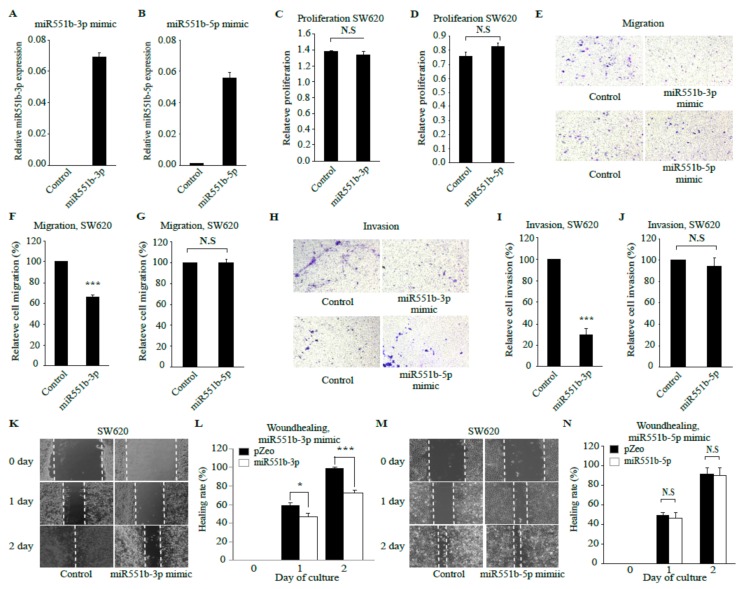
**miR551b-3p mimic, but not miR551b-5p mimic,****suppresses the motility of CRC cell lines.** (**A** and **B**), SW620 cells were transiently transfected with either control, miR551b-3p mimic, or miR551b-5p mimic, and the expression levels of miR551b-3p and miR551b-5p were determined by RT-qPCR. (**C** and **D**), SW620 cells transfected with either control, miR551b-3p mimic, or miR551b-5p mimic were incubated for 72 h to determine the proliferation rate by MTT assay. N.S means not significant. (**E**–**J**), SW620 cells transfected with either control, miR551b-3p mimic, or miR551b-5p mimic were seeded in a transwell non-coated or coated with Matrigel, followed by incubation for 24 h for evaluating cell migration and invasion, respectively. Images were obtained using an inverted microscope. Representative images were shown (**E**, migration; **H**, invasion). Data were presented as mean ± SEM of three independent experiments (**E**–**G**, migration; **H**–**J**, invasion). *** *p* < 0.001. N.S means not significant. (**K**–**N**), SW620 cells transduced with control, miR551b-3p mimic, or miR551b-5p mimic were seeded and analyzed by to in vitro scratch assays. Images were captured at 0, 1, and 2 days after incubation. The dotted lines defined the areas lacking cells. Representative images were shown (**K** and **M**). Data are presented as mean ± SEM of three independent experiments. * *p* < 0.05, *** *p* < 0.001. N.S means not significant.

**Figure 4 cancers-11-00735-f004:**
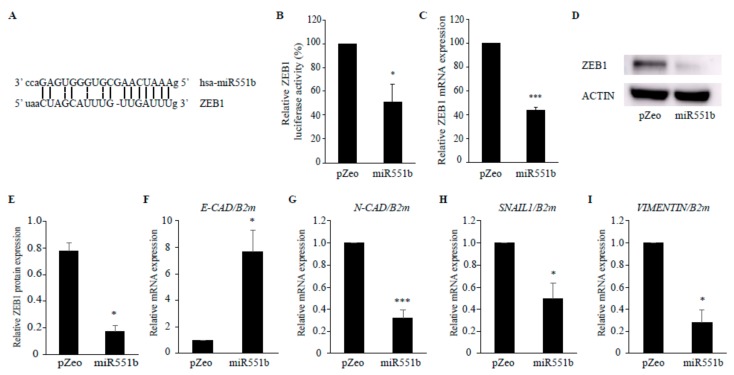
**ZEB1 was a direct target of miR551b.** (**A**), Sequence alignment of miR551b with the 3′ UTR of ZEB1 binding site was shown. (**B**), 293T cells were transfected with either control vector or miR551b together with psiCheck2; samples were analyzed by dual luciferase reporter assay. (**C**–**E**), ZEB1 expression was evaluated in control (pZeo)- or miR551b- transduced SW620 cells by RT-qPCR (**C**) and immunoblotting (**D**,**E**). ACTIN was used as the loading control. Representative images were shown (**D**). Relative protein expression of ZEB1 was quantified using Image J (**E**). Data were presented as mean ± SEM of three independent experiments. * *p* < 0.05, *** *p* < 0.001. (**F**–**I**), RNA was isolated from control vector (pZeo)- or miR551b-transduced SW620 cells; expression levels of *E-CAD* (**F**), *N-CAD* (**G**), *SNAIL* (**H**), and *VIMENTIN* (**I**) were determined by RT-qPCR. The data shown were mean ± SEM of three independent experiments. * *p* < 0.05, *** *p* < 0.001.

**Figure 5 cancers-11-00735-f005:**
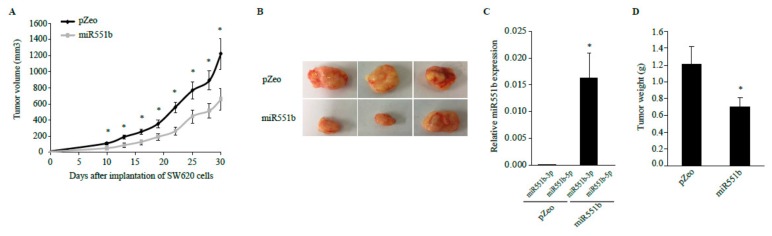
**miR551b suppressed the xenograft of SW620 cells in vivo.** (**A**), Control (pZeo)- or miR551b- transduced SW620 cells were subcutaneously injected into NSG mice. The mice were then monitored for tumor growth. Tumor volume was measured every 3 days up to 1 month (*n* = 6). Data were presented as mean ± SEM. * *p* < 0.05. (**B**), Representative images of tumors were shown. (**C**), RNA was isolated from each tumor, and miR551b-3p and miR551b-5p expression levels were analyzed by RT-qPCR. The data shown were mean ± SEM (*n* = 5). * *p* < 0.05. (**D**), The tumors were isolated from each mouse and weighed. Data were presented as mean ± SEM (*n* = 6). * *p* < 0.05.

**Figure 6 cancers-11-00735-f006:**
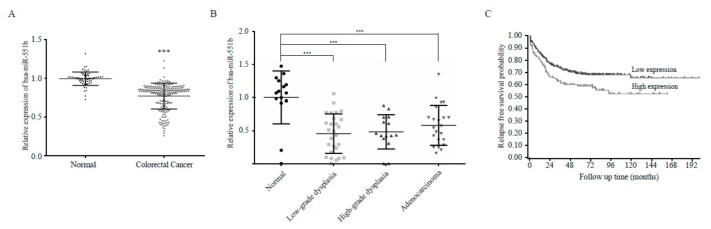
**miR551b is inversely correlated with progression of CRC patients.** (**A**) The expression level of hsa-miR-551b in normal and CRC patient samples was analyzed using microRNA datasets GSE81582, GSE41655, GSE18392, GSE98406, and GSE30454 (normal, *n* = 64; CRC, *n* = 208; mean ± s.e.m.). *** *p* < 0.001. (**B**) Dataset GSE41655 was analyzed to determine the correlation of hsa-miR-551b expression with CRC stage (normal, *n* = 15; low-grade dysplasia *n* = 27; high-grade dysplasia *n* = 15; adenocarcinoma, *n* = 22). *** *p* < 0.001 by one-way ANOVA. (**C**) Kaplan-Meier curves of relapse-free survival probability in CRC patients (GSE39582) based on the expression of ZEB1 (high, *n* = 140; low, n = 417). *p* = 0.01.

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
