# Peer review of "miR551b Regulates Colorectal Cancer Progression by Targeting the ZEB1 Signaling Axis"

_cancers, 2019, doi:10.3390/cancers11050735_

Reviewer 1 Report

Authors have showed the role of miR-551b-3p in colorectal cancer progression. Author concluded that miR-551b-3p ectopic expression targets ZEB1 and showed decrease in malignant phenotypes and tumor growth of CRC. Authors have performed the study clearly. There are few concerns that needs to be addressed:

Please mention selection method for positive miR551b cells after infection with pZeo vector in line 214 or in Materials and Methods.

Please provide the details of more studies in Introduction to support miR551b as a tumor suppressor and targets proto-oncogenes.

As authors have cited good quality of journals to support the study, please change the reference no. 23 also with a journal which have good impact factor.

Figure 2I doesn’t show any standard error. Please explain as invasion assay show some changes in triplicates value.

Author showed wound complete closure with SW620 control in 3K while not in 3M. Until 1 day, wound closure seemed equal in both, while it is different in 2nd day. Please explain.

In discussion, please mention about published studies which show different microRNAs targeting ZEB1 in CRC or any other cancer.

Author Response

Response to the comments from reviewers

Reviewer #1 (Comments to the Author): 

Authors have showed the role of miR-551b-3p in colorectal cancer progression. Author concluded that miR-551b-3p ectopic expression targets ZEB1 and showed decrease in malignant phenotypes and tumor growth of CRC. Authors have performed the study clearly. There are few concerns that needs to be addressed:

1)  Please mention selection method for positive miR551b cells after infection with pZeo vector in line 214 or in Materials and Methods.

In response to the comment, we edited the methods as shown below.

After lentiviral infection, GFP positive cells were sorted by FACS Aria III (BD Bioscience, USA) and used for experiments.

2)  Please provide the details of more studies in Introduction to support miR551b as a tumor suppressor and targets proto-oncogenes.

We added one more reference, a total of 5, which are all the references that are available currently. Also, we edited the manuscript as shown below.

By contrast, miR551b-3p expression levels were found to be inversely correlated with the prognosis of gastric cancer (GC) as a result of inhibiting ERBB4 expression [22,23]. The expression of miR551b was much lower in gastric tumors than in normal tissues, and miR551b mimics significantly decreased the proliferation and invasion of gastric cancer cell line SGC-7901 [24].

3)  As authors have cited good quality of journals to support the study, please change the reference no. 23 also with a journal which have good impact factor.

We agree with the reviewer’s comment, but the three references (22,23,24) that we cited are the only references available currently.

4) Figure 2I doesn’t show any standard error. Please explain as invasion assay show some changes in triplicates value.

We are sincerely sorry for the mistake. We found that the error bar was deleted while making the figure. Thus, we replaced it with an edited one.

5)  Author showed wound complete closure with SW620 control in 3K while not in 3M. Until 1 day, wound closure seemed equal in both, while it is different in 2nd day. Please explain.

The figure 3L and 3N, showing the average data of wound closure of 3K and 3M, indicate about 95 to 99% of wound closure of pZeo control at 2nd day. Though 3K and 3M look a little different, we would say that they are very similar by average. If wanted, we will replace it.

6)   In discussion, please mention about published studies which show different microRNAs targeting ZEB1 in CRC or any other cancer.

We added more references and edited the manuscript as shown below.

Several miRNAs, such as miR200 [30], miR1199-5p [31], and miR128-3p [32], were shown to target ZEB1, leading to the coordination of EMT

Reviewer 2 Report

The research article proposed by Kim et al. has an approach to the clinical question valid and the study design (number of clinical sample, animal model and molecular analysis performed) are well  planned and sufficiently complete. Finally, the article is informative, the results fit with the initial hypothesis and the conclusions supported by the results.

Nevertheless, I have the following suggestion that would strengthen the article:

a)    In the introduction section, page 1, line 31, the authors refer to the epidemiological data that date back to 2012. However, recently has been published the GLOBOCAN statistics updated to 2018 statistics: Bray F, Ferlay J, Soerjomataram I, Siegel RL, Torre LA, Jemal A. Global cancer statistics 2018: GLOBOCAN estimates of incidence and mortality worldwide for 36 cancers in 185 countries. CA Cancer J Clin 2018;68:394-424. Please change the data with the updated statistics.

b)    In the introduction section, page 1, line 43, in order to reinforce the concept should consider to cite the following paper: Front Biosci (Schol Ed). 2014 Jan 1;6:110-9.

c)    In Material and Methods section, page 2, line 85, in order to make the experiments reproducible, the authors must indicate the size of the transwell pores.

d)    In Material and Methods section, page 3, line 99-104, why do the authors use three different RNA extraction methods, two different cDNA retrotranscription methods and two different qRT-PCR kit? Please specify the plausible reasons.

e)    In Material and Methods section, page 3, line 113-118, the authors for western blot analysis used five different antibody. In order to make the experiments reproducible and summarize everything clearly have to create a table (or alternatively a supplementary table) in which they summarize the name of protein recognized, the host, the dilution used and the producers’ name.

f)     In result section, page 4, line 173-174, the authors refer to a patient-derived samples in which quantify the relative expression of miR551-3p/-5p. However, there is no trace of the characteristics of these patients in Material and Methods section. How many patients enrolled? How many normal and tumor samples are? What are their clinical pathological characteristics? The number of patient or samples is indicated in the figure legend 1 but is not enough. Please add a table in which summarize all the patients clinical pathological characteristics. In addition, my major concern is the very low number of patient enrolled (n=4). The cohort of patients’ needs to be improved.

g)    In result section, page 8, line 261-264, the following sentence: “To test this, the ZEB1 3’-UTR was cloned into the psiCheck-2 vector, after which 261 293T cells were transiently transfected with either control vector and ZEB1 UTR or miR551b 262 overexpression and ZEB1 UTR. Luciferase reporter assay was then performed to determine the 263 binding of miR551b on the ZEB1 UTR.”, is a repetition of what is indicated in the materials and methods and must be deleted.

h)    In result section, page 8, line 275-278, the authors affirm that since miR551 is deregulated in CRC and interact with 5’-UTR of ZEB1 it is “EXPECTED” that also all other genes of EMT (E-cad, N-cad, Vimentin and Snail) are deregulated. Is this really expected? Why? Can the authors better explain which are the direct/indirect mechanisms that lead to a general deregulation of the EMT pathway following the interaction of miR551 with ZEB1?

i)      In discussion section, page 10, line 337-338, the abbreviation of ”high-risk head and neck cancer” and “gastric cancer” have already been used since page 2, no need to repeat them every time. Just use directly the abbreviation. The same for “ovarian cancer” used here for the first time.

Author Response

Response to the comments from reviewers.

Reviewer #2 (Comments to the Author): 

The research article proposed by Kim et al. has an approach to the clinical question valid and the study design (number of clinical sample, animal model and molecular analysis performed) are well planned and sufficiently complete. Finally, the article is informative, the results fit with the initial hypothesis and the conclusions supported by the results.

Nevertheless, I have the following suggestion that would strengthen the article:

a)  In the introduction section, page 1, line 31, the authors refer to the epidemiological data that date back to 2012. However, recently has been published the GLOBOCAN statistics updated to 2018 statistics: Bray F, Ferlay J, Soerjomataram I, Siegel RL, Torre LA, Jemal A. Global cancer statistics 2018: GLOBOCAN estimates of incidence and mortality worldwide for 36 cancers in 185 countries. CA Cancer J Clin 2018;68:394-424. Please change the data with the updated statistics.

In response to the comment, we edited it in the introduction part as follows.

Colorectal cancer (CRC) is the third most commonly diagnosed cancer and second leading cause of cancer-related deaths worldwide, accounting for approximately 1.8 million new cases and about 881,000 deaths for both sex in 2018 [1].

b)  In the introduction section, page 1, line 43, in order to reinforce the concept should consider to cite the following paper: Front Biosci (Schol Ed). 2014 Jan 1;6:110-9.

We checked the reference as suggested, but, in our view, it was context-wise difficult to fit in the sentence. We hope that reviewer would check it one more time and let us know if it is really necessary.

c)  In Material and Methods section, page 2, line 85, in order to make the experiments reproducible, the authors must indicate the size of the transwell pores.

In response to the comment, we put the transwell insert pore size, as shown below.

The transwell insert (8μm pore size) system (Corning, USA) coated with or without 20 μl of Matrigel (BD, USA) were respectively used to examine the cell invasion and migration in vitro as shown before [27].

d)  In Material and Methods section, page 3, line 99-104, why do the authors use three different RNA extraction methods, two different cDNA retrotranscription methods and two different qRT-PCR kit? Please specify the plausible reasons.

       We isolated and generated cDNA from total RNA and miRNA with different kits because the method of isolating total RNA and miRNA are different. 

e)  In Material and Methods section, page 3, line 113-118, the authors for western blot analysis used five different antibody. In order to make the experiments reproducible and summarize everything clearly have to create a table (or alternatively a supplementary table) in which they summarize the name of protein recognized, the host, the dilution used and the producers’ name.

We are sorry for the mistake. Actually, it was not necessary part so that we deleted the part.

f)  In result section, page 4, line 173-174, the authors refer to a patient-derived samples in which quantify the relative expression of miR551-3p/-5p. However, there is no trace of the characteristics of these patients in Material and Methods section. How many patients enrolled? How many normal and tumor samples are? What are their clinical pathological characteristics? The number of patient or samples is indicated in the figure legend 1 but is not enough. Please add a table in which summarize all the patients clinical pathological characteristics. In addition, my major concern is the very low number of patient enrolled (n=4). The cohort of patients’ needs to be improved.

In response to the comment, we increased the sample number from 4 to 10 pairs, and edited the manuscript as suggested. Also, the clinical pathological characteristics of patients were summarized in methods section and supplementary table2 as shown below.

      2.13. Information of tissue specimens 

      For miRNA qRT-PCR experiments, the 10 pair tissues were obtained from colorectal carcinoma patients who underwent surgery at Soonchunhyang          University Cheonan Hospital in South Korea. All clinicopathological data including age, gender, TNM stage, vascular invasion, Lymphatic invasion, and perineural invasion were noted in Supplementary Table 2. This study and clinicopathological data were approved by the Ethics Committee of         Soonchunhyang University, Cheonan Hospital (2018-07-061-003).

g)  In result section, page 8, line 261-264, the following sentence: “To test this, the ZEB1 3’-UTR was cloned into the psiCheck-2 vector, after which 261 293T cells were transiently transfected with either control vector and ZEB1 UTR or miR551b 262 overexpression and ZEB1 UTR. Luciferase reporter assay was then performed to determine the 263 binding of miR551b on the ZEB1 UTR.”, is a repetition of what is indicated in the materials and methods and must be deleted.

In response to the comment, we deleted that sentence in the result part.

h)  In result section, page 8, line 275-278, the authors affirm that since miR551 is deregulated in CRC and interact with 5’-UTR of ZEB1 it is “EXPECTED” that also all other genes of EMT (E-cad, N-cad, Vimentin and Snail) are deregulated. Is this really expected? Why? Can the authors better explain which are the direct/indirect mechanisms that lead to a general deregulation of the EMT pathway following the interaction of miR551 with ZEB1?

We edited the sentence and put more references which show that ZEB1 inhibits the expression of E-cadherin as shown below.

Epithelial gene E-CAD level was upregulated following miR551b overexpression, whereas the levels of mesenchymal genes, such as N-CAD, SNAIL, and VIMENTIN, were downregulated by miR551b overexpression (Fig. 4F-I), suggesting that miR551b overexpression leads to the dysregulation of EMT signatures in CRC cells by targeting ZEB1. These data were consistent with the previous reports, showing that ZEB1 is implicated in EMT by repressing E-CAD [28-31].

i)  In discussion section, page 10, line 337-338, the abbreviation of ”high-risk head and neck cancer” and “gastric cancer” have already been used since page 2, no need to repeat them every time. Just use directly the abbreviation. The same for “ovarian cancer” used here for the first time.

In response to the comment, we changed the sentence.